# Joint Characterization and Fractal Laws of Pore Structure in Low-Rank Coal

Yuxuan Zhou [1,2,*], Shugang Li [1,2,*], Yang Bai [1,2], Hang Long [1,2], Yuchu Cai [1,2] and Jingfei Zhang [1,2]

[1] School of Safety Science and Engineering, Xi'an University of Science and Technology, Xi'an 710054, China
[2] Key Laboratory of Coal Resources Exploration and Comprehensive Utilisation, Ministry of Natural Resources, Xi'an 710054, China
* Correspondence: zyx@stu.xust.edu.cn (Y.Z.); lisg@xust.edu.cn (S.L.)

**Abstract:** The pore structure of low-rank coal reservoirs was highly complex. It was the basis for predicting the gas occurrence and outburst disasters. Different scale pores have different effects on adsorption–desorption, diffusion, and seepage in coalbed methane. To study the pore structure distribution characteristics, which are in different scales of low-rank coal with different metamorphism grade, the pore structure parameters of low-rank coal were obtained by using the mercury injection, $N_2$ adsorption, and $CO_2$ adsorption. These three methods were used to test the pore volume and specific surface area of low-rank coal in their test ranges. Then, the fractal dimension method was used to calculate the fractal characteristics of the pore structure of full aperture section to quantify the complexity of the pore structure. The experimental results showed that the pore volume and specific surface area of low-rank coal were mainly controlled by microporous. The pore fractal characteristics were obvious. With the influence of coalification process, as the degree of coal metamorphism increases, fluctuations in the comprehensive fractal dimension, specific surface area, and pore volume of the pore size range occur within the range of $R_{max} = 0.50\%$ to $0.65\%$.

**Keywords:** low-rank coal; pore structure; comprehensive fractal dimension; fractal law





## 1. Introduction

Coal bed methane (CBM), as clean energy, will promote the sustainable development of global energy supplying [1]. The reserves of CBM will depend on the safety and reliability of national energy. Low-rank coal is of complex pore structure, especially rich in microporous, which control the adsorption capacity of CBM, This is the basis for evaluating coal mine disasters [2–4]. Therefore, it is of great significance to identify the distribution of the pores about low-rank coal.

The pore structure plays a crucial role in various fields such as coal mining, gas extraction, and hazard prediction [5–7]. The size and connectivity of pores directly impact the permeability and extraction rate of gas [8]. Larger pores and improved connectivity facilitate enhanced gas permeability and extraction efficiency while reducing gas pressure [9]. Simultaneously, larger pore surfaces provide more adsorption sites for gas adsorption. This increases the chances of gas molecules coming into contact with the pore walls, thereby enhancing gas adsorption capacity [10]. The pore structure also influences the flotation kinetics of coal. It affects the diffusion rate and adsorption rate of the flotation reagents, as well as the stability of the foam, thereby impacting the flotation speed and recovery rate of coal [11]. By conducting ventilation borehole monitoring of gas content within coal seams, it becomes possible to assess the gas composition and concentration and predict potential hazards associated with the pore structure [12] because pores can be used as gas storage and migration channels. When a large amount of gas accumulates in the coal seam, larger pores and better connectivity help to improve gas permeability [13]. However, at the same time, it will increase the speed and scale of gas accumulation, thus increasing the risk of gas explosion [14]. Secondly, the mining disturbance will make the coal dust

suspended in the air, and the fire source may cause the coal dust explosion. The pore size and distribution characteristics directly affect the suspension capacity of coal dust and the risk of coal dust explosion. Smaller pores may be more likely to suspend coal dust and form explosive dust clouds, increasing the aerodynamic risk during mining [15]. Therefore, the study of pore structure plays an irreplaceable role throughout the entire process of coal mining and utilization.

There are many experimental methods for testing the pore structure, including gas adsorption, visualization, and capillary pressure methods [16]. The pore structure characteristics of coal are mainly tested using scanning electron microscopy [17], gas adsorption [18,19], mercury injection (MIP) [20], nuclear magnetic resonance (NMR) [21,22], X-ray diffraction (XRD) [23], CT imaging (X-CT) [24], and small angle scattering (SAXS) [25]. The pore structure has been studied by many scholars using the above means. Using scanning electron microscopy, XRD, etc., the pore morphology and connectivity of coal rock bodies can be obtained visually [26–29]. However, the pore characteristic data observed by these methods are mainly used for qualitative characterization, and it is difficult to analyze quantitatively.

To obtain a quantitative characterization of the pore structure, mercury injection, gas adsorption, and small-angle scattering methods were introduced. Obtaining pore size parameters through fluid intrusion into the pore of coal bodies is widely recognized because of their wide measurement range and high measurement accuracy [30–32]. The low-temperature $N_2$ adsorption method and low-pressure $CO_2$ adsorption method were employed to measure the characteristics of mesopores and micropores, respectively [33]. After obtaining the pore structure parameters, the introduction of fractal dimension calculation provides a quantitative characterization of the complexity of the pore network [34].

Due to the limitations of the type of pore structure testing method, different experiments could only characterize the pore distribution at a certain scale but cannot fully reflect the pore structure characteristics. In recent years, scholars have successively attempted to combine multiple testing methods to jointly characterize the pore structure features and pore distribution of coal [35–38]. However, some scholars have only tested mesopores and macropores [39], while others have not quantified the complexity of the pore structure [40,41].

This study is to complete a more comprehensive joint characterization of a full aperture about the pore structure in low-rank coal and to quantify the pore structure complexity. Eight low-rank coal samples from the northern Shaanxi coalfield were used to investigate the pore joint characterization. Pore structure characteristics are evaluated using mercury injection, low-temperature $N_2$ adsorption, and $CO_2$ adsorption, to study the causes of pore structure complexity in low-rank coal. The study had significant implications for CBM reservoir and transport patterns.

## 2. Materials and Methods

### 2.1. Sample Preparation

Primary coal from eight different mines in the northern Shaanxi coalfield were selected for the experiments, from Bailiang 5# coal seam (BL), Liangshuijing 4-3 coal seam (LSJ), Daljuta (DLT), Ningtiaota 3-1 coal seam (NTT), Jianxin 4-2 coal seam (JX), Ruineng 401 coal seam (RN), Xiaozhuang 4# coal seam (XZ) and Huangling 2# coal (HL). The distribution of coal mines was shown in Figure 1.

The samples were collected at the newly exposed coal wall, sealed, and brought back to the laboratory. The coal samples were crushed by grinding, and 2.4–4 mm coal samples were screened for mercury injection. In addition, 0.18–0.25 mm coal samples were screened for low-temperature $N_2$ adsorption and carbon dioxide adsorption experiments. An amount of 10 g of each sample was weighed to set aside. The experimental coal samples were analyzed according to ISO 11722:2013 [42] *(Solid mineral fuels—Hard coal—Determination of moisture in the general analyzed test sample by drying in nitrogen)* and ISO 1171:2010 [43] *(Solid mineral fuels-Determination of ash)*, and the maximum reflectance of

the specular group was determined according to the national standard GBT6948-2008 [44] *(Method of determining microscopically the reflectance of vitrinite in coal)*. The test results were shown in Table 1.

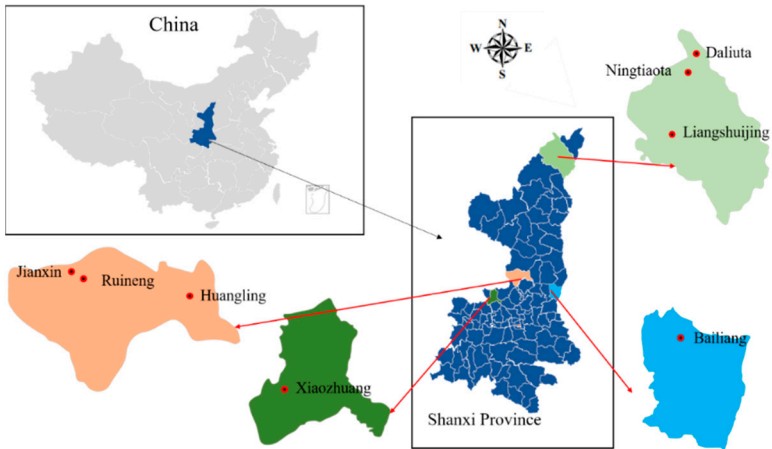

**Figure 1.** Distribution map of mining areas.

**Table 1.** Results of coal quality analysis.

| Sample ID | Proximate Analysis (%) | | | | $R_{max}$/% |
|---|---|---|---|---|---|
| | $M_{ad}$ | $A_{ad}$ | $V_{daf}$ | $FC_{ad}$ | |
| BL | 0.81 | 27.36 | 12.72 | 59.43 | 0.33 |
| LSJ | 4.55 | 4.69 | 30.73 | 61.64 | 0.47 |
| DLT | 6.89 | 6.95 | 31.49 | 57.32 | 0.49 |
| NTT | 5.26 | 4.14 | 32.76 | 59.78 | 0.53 |
| JX | 2.94 | 9.05 | 36.66 | 52.70 | 0.55 |
| RN | 2.46 | 13.7 | 30.86 | 54.08 | 0.60 |
| XZ | 3.90 | 5.74 | 33.08 | 58.79 | 0.66 |
| HL | 2.1 | 4.86 | 31.03 | 62.76 | 0.68 |

$M_{ad}$ is Moisture, $V_{daf}$ is Volatile fraction, $A_{ad}$ is Ash, $FC_{ad}$ is Fixed carbon, and $R_{max}$ is Maximum vitrinite reflectance.

The low-rank coal samples with volatile matter were ranging from 12.72% to 36.66% (10% to 40% volatile matter for low-rank coal) [45]. The specular group emissivity characterized the degree of deterioration of the coal samples (maximum specular group reflectance $R_{max} < 0.68\%$ for low-rank coal), the lower degree of coal deterioration with the lower the $R_{max}$.

## 2.2. High-Pressure Mercury Injection Experiments

The pore structure of the experimental coal samples was analyzed using an AutoProe IV 9510 fully automatic mercury injection [46,47]. The test system diagram is shown in Figure 2a.

Coal samples were prepared by weighing about 3 g, and it was dried at 70 °C for 8 h and vacuumed in the dilatometer. The experimental temperature was 298 K, the maximum working pressure was 414 MPa, and testable pore size ranges from 3 nm to 1000 μm. During high-pressure mercury injection experiments, the native pore structure of the coal sample was damaged as the mercury solution was pressed in. The experiment had a greater advantage for mesoporous and macropore tests but a greater deviation for microporous and mesoporous tests. The specific test conditions were as follows: in/out mercury contact angle 130°, surface tension 0.48 N/m, and expansion gauge volume 0.5 cc. The relationship between inlet pressure and pore size can be obtained using the Washburn equation [48]. The pore surface area parameter can be determined using the theoretical model proposed by Rootare [49].

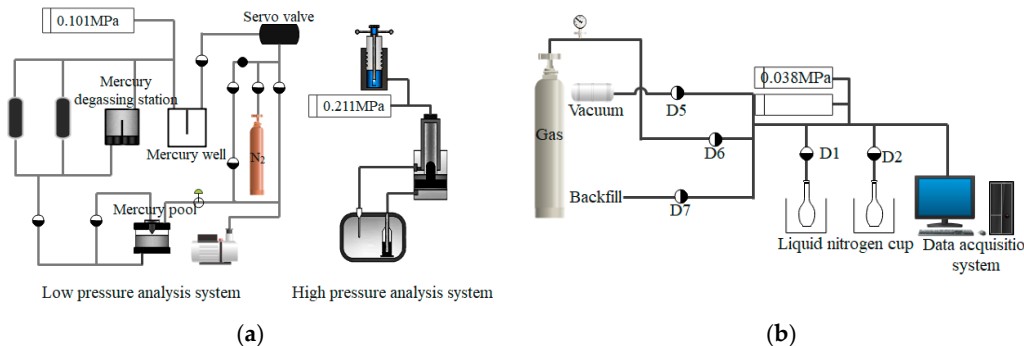

<div align="center">(<b>a</b>)                                    (<b>b</b>)</div>

**Figure 2.** Experimental system. (**a**) High-pressure mercury injection experiments; (**b**) Low-temperature $N_2$ adsorption and carbon dioxide adsorption experiments.

### 2.3. Low Temperature $N_2$ Adsorption Experiments and Low Pressure $CO_2$ Adsorption Experiments

The pore structure of experimental coal samples was analyzed by ASAP 2460 specific surface area and porosity tester [50]. The test system diagram is shown in Figure 2b. The maximum working pressure is 133 MPa, and pore size test ranges from 0.35 nm to 500 nm. The prepared coal samples were weighed to approximately 200 mg and dried at 70 °C for 8 h before the experiments were carried out.

Low-temperature nitrogen adsorption experiments are carried out at liquid nitrogen temperature (77 K). The amount of nitrogen adsorbed on the solid surface depends on the relative pressure of the nitrogen ($P/P_0$). $P$ is partial pressure, and $P_0$ is saturation steam pressure. When $P/P_0$ is between 0.05 and 0.35, the relationship between adsorption and relative pressure is in accordance with the BET equation, and it is the basis for the determination of the specific surface area of powder materials by low-temperature nitrogen adsorption. When $P/P_0 \geq 0.40$, nitrogen agglomerates in microporous. The parameters such as pore volume, pore size distribution, and specific surface area can be determined according to BJH or DFT theoretical models.

Low-pressure $CO_2$ adsorption experiments were carried out at saturation temperature (273 K). Testable pore sizes in the range of less than 2 nm. The principle of carbon dioxide adsorption experiments was similar to low-temperature $N_2$ adsorption experiments. However, carbon dioxide molecules are smaller and diffuse at a faster rate, it had a greater saturation pressure at saturation temperature and can be tested on micropore.

In order to ensure the accuracy of the experimental data, each group of experiments were tested three times. The final data used were the average of the three groups of experiments. The relative error of each group is less than 5%.

## 3. Results and Discussion

### 3.1. Adsorption Curves and Pore Distribution Characteristics

The article used the IUPAC pore classification method. Pore was classified as micropore (<2 nm), mesopore (2 nm to 50 nm), and macropore (>50 nm).

The mercury injection experiments focused on the pore characteristics of the mesoporous and macropore. The mercury intrusion and exit curves are drawn based on the experimental parameters of coal samples (Figure 3).

The opening degree of pore in coal samples could be reflected by different curves with mercury entry and exit. All eight low-rank coal samples showed significant mercury injection hysteresis loops, indicating that the open-pore spaces are more developed and the inter-pore connectivity is better. The efficiency of mercury exit was calculated to be 30.85–48.34%. The mercury exit efficiency is medium, indicating that there are both open pore and semi-open pore in coal samples, and the pore connectivity is good. Mercury exit efficiency of low-rank coal is less than 40% with $R_{max} = 0.60\%$ and $R_{max} < 0.50\%$. And there were in the range of 40–50% with $R_{max} = 0.50$–$0.60\%$ and $R_{max} > 0.60\%$. The calculations show that the pore connectivity and openness of the more highly metamorphosed low-rank coal are better than less metamorphosed low-rank coal.

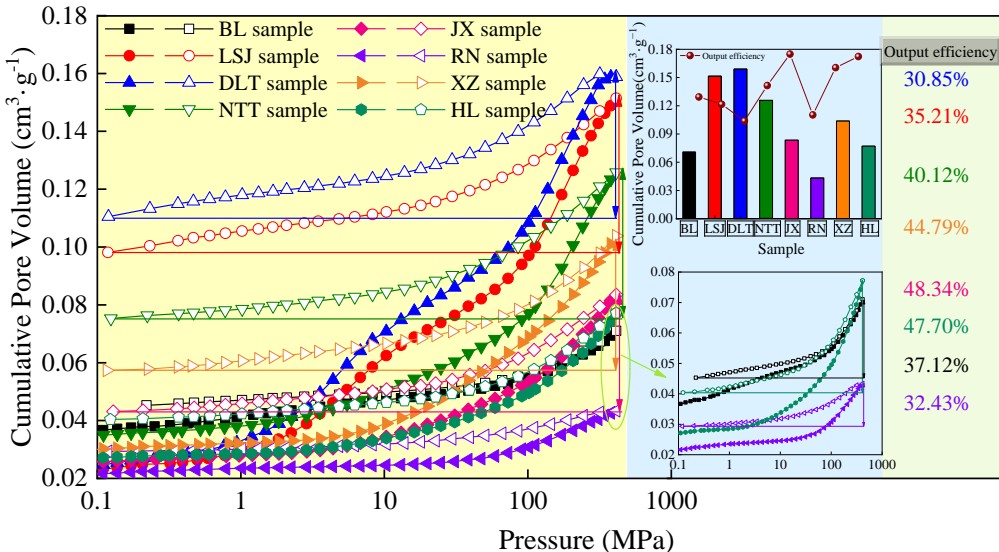

**Figure 3.** Mercury entry and withdrawal curves of the mercury intrusion experiments.

When the pressure is greater than 10 MPa, the compression effect of the coal matrix is obvious [51,52]. Affected by this effect, the reflection of the real pore structure is biased. With the increase of pressure, the deformation is more obvious, and the deformation of pore under this pressure must be considered [53,54]. It is necessary to modify the experimental data of mercury injection with pressure above 10 MPa, and the corresponding pore size is about 135 nm.

$$\begin{cases} V_{xPi} = V_{cPi} - ZV_m(P_i)P_i \\ V_m(P_i) = V_m - \dfrac{dV_{cPi}}{dP_i}P_i \\ Z = \dfrac{1}{V_m}\left(K - \dfrac{\Delta V_P}{\Delta P}\right) \end{cases} \tag{1}$$

where $P_i$ is the mercury inlet pressure corresponding to the pore volume to be modified (>10 MPa), MPa; $V_{Pi}$ is the hole volume test value corresponding to attack pressure $P_i$, cm³/g; $V_{XPi}$ is the corrected pore volume corresponding to the inlet mercury pressure $P_i$, cm³/g; $V_m(P_i)$ is the coal matrix volume after the change of mercury injection pressure, cm³/g; $V_m$ is the volume of coal matrix, cm³/g; Z is compression coefficient of coal matrix; $\Delta V_P$ is the cumulative pore volume value of mercury entry section to be corrected, cm³/g; $\Delta P$ is the difference between the maximum value and the minimum value of the mercury injection pressure to be corrected, MPa; K is the slope of the high-pressure stage (>10 MPa).

The curves were obtained by fitting a linear regression to the mercury feed curve (Figure 4).

The pore volume and specific surface area data were corrected for the high-pressure section of the mercury injection experiment (Table 2).

**Table 2.** Correction results for mercury injection experimental data.

| Simple ID | K | V/cm³·g⁻¹ | $V_X$/cm³·g⁻¹ | S/cm³·g⁻¹ | $S_X$/cm³·g⁻¹ |
|---|---|---|---|---|---|
| BL | $1.300 \times 10^{-4}$ | 0.0024 | 0.0015 | 0.1750 | 0.2828 |
| LSJ | $7.694 \times 10^{-4}$ | 0.0153 | 0.0045 | 1.0897 | 3.6836 |
| DLT | $8.477 \times 10^{-4}$ | 0.0154 | 0.0036 | 1.1764 | 4.9916 |
| NTT | $5.225 \times 10^{-4}$ | 0.0095 | 0.0029 | 0.5989 | 1.9913 |
| JX | $3.903 \times 10^{-4}$ | 0.0069 | 0.0013 | 0.4157 | 2.2026 |
| RN | $6.259 \times 10^{-4}$ | 0.0010 | 0.0001 | 0.0642 | 0.0647 |
| XZ | $5.782 \times 10^{-4}$ | 0.0100 | 0.0036 | 0.5840 | 1.6372 |
| HL | $2.952 \times 10^{-4}$ | 0.0053 | 0.0020 | 0.3212 | 0.8500 |

K is the experimental result for pore volume of pore > 50 nm in diameter, $V_X$ is the corrected pore volume for pore > 50 nm in diameter, S is the experimental result of the specific surface area of pore with a pore size > 50 nm, $S_X$ is the specific surface area after correction for pore size > 50 nm pore.

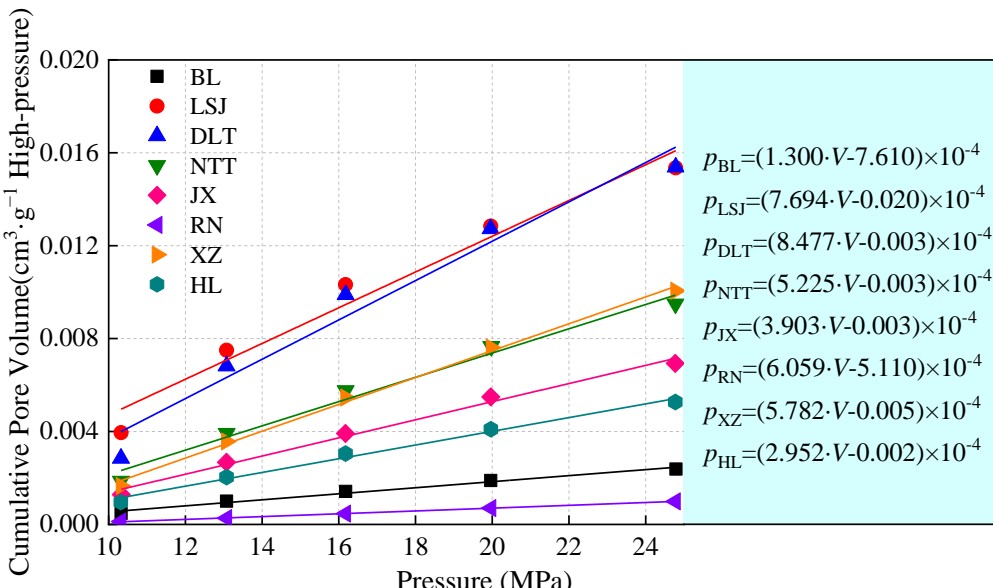

**Figure 4.** Linear regression of the high-pressure section of the mercury injection curve.

Low-temperature nitrogen adsorption experiments are better for mesopore [55]. The adsorption isotherms were plotted from the low-temperature $N_2$ adsorption experimental data for each coal sample (Figure 5). LSJ, DLT, and NTT coal samples at the start of the low-pressure section with the curve were biased towards the Y axis. These three coal samples had a strong interaction with liquid nitrogen, with more microporous present in this category and a strong adsorption potential within the microporous. The starting segment was type I according to the IUPAC isotherm classification. The starting adsorption curve of BL, JX, RN, XZ, and HL coal samples was type III, which had a weak interaction force with liquid nitrogen.

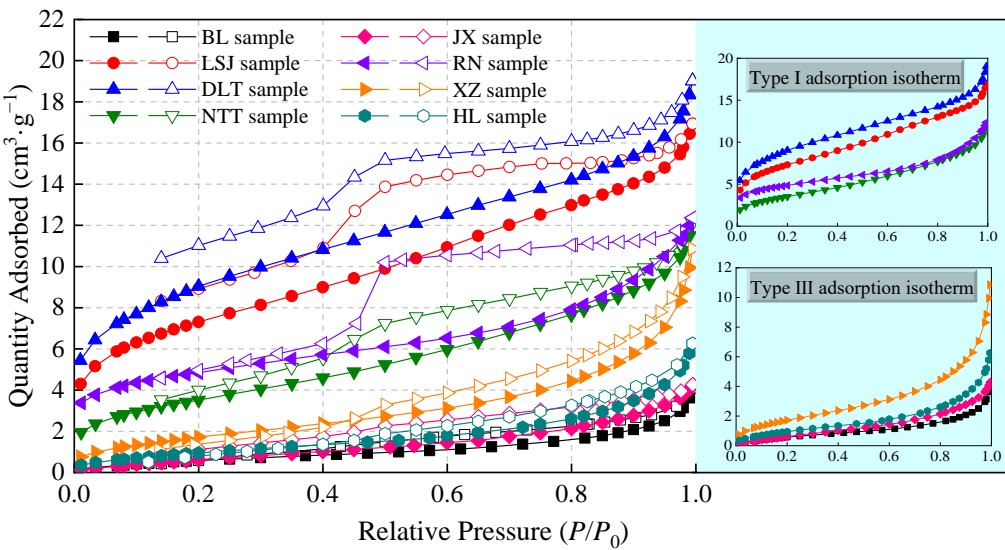

**Figure 5.** Low-temperature nitrogen adsorption isotherms.

$CO_2$ adsorption experiments are more capable of testing microporous. In the microporous stage, $CO_2$ was presented as a monolayer adsorption or microporous filling on the coal surface. Therefore, the adsorption and desorption curves overlap (Figure 6). The shape of the isothermal sorption curve for low-rank coal samples was fitted by the Langmuir equation. The adsorption volume increased rapidly, and the adsorption curve is clearly raised upwards in the low relative pressure region. It tends to be straight in the

high-pressure region. The adsorption capacity shows a general trend of decreasing with increasing coal deterioration, except for the BL, XZ, and HL coal samples.

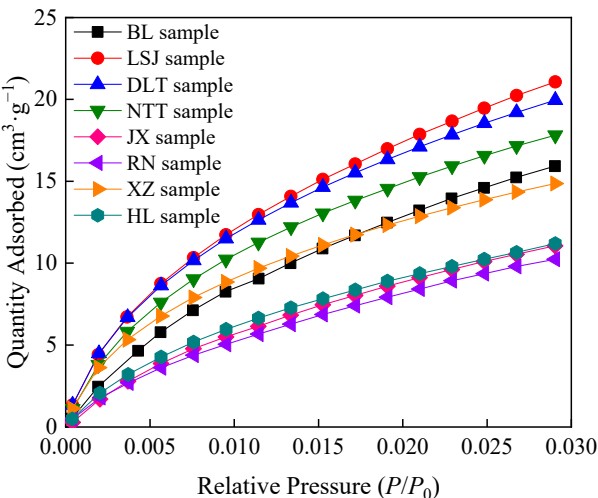

**Figure 6.** $CO_2$ adsorption isotherms.

The pore structure parameters tested by mercury pressure, low-temperature $N_2$ adsorption, and $CO_2$ adsorption experiments are shown in Table 3.

**Table 3.** Test parameters for mercury injection, low-temperature $N_2$ adsorption, and low-pressure $CO_2$ adsorption experiments.

| | Sample ID | BL | LSJ | DLT | NTT | JX | RN | XZ | HL |
|---|---|---|---|---|---|---|---|---|---|
| Mercury injection experiment | Porosity/% | 8.39 | 15.81 | 16.31 | 13.37 | 9.33 | 6.62 | 11.28 | 8.47 |
| | Average pore size/nm | 19.90 | 14.10 | 16.80 | 12.90 | 12.90 | 16.70 | 13.40 | 12.80 |
| | Pore volume/cm$^3$·g$^{-1}$ | 0.0710 | 0.1515 | 0.1588 | 0.1257 | 0.0835 | 0.0434 | 0.1038 | 0.0772 |
| | Pore specific surface area/m$^2$·g$^{-1}$ | 14.2830 | 42.9030 | 37.8090 | 39.0020 | 25.8020 | 10.3630 | 31.069 | 24.1220 |
| Low pressure nitrogen adsorption experiment | Average pore size/nm | 11.08 | 4.85 | 4.56 | 6.43 | 7.80 | 6.64 | 10.57 | 10.21 |
| | BJH pore volume/cm$^3$·g$^{-1}$ | 0.0059 | 0.0223 | 0.0239 | 0.0172 | 0.0063 | 0.0153 | 0.0160 | 0.0092 |
| | BET pore specific surface area/m$^2$·g$^{-1}$ | 2.5756 | 25.8364 | 31.9735 | 12.8436 | 3.1947 | 16.6833 | 6.6900 | 3.9026 |
| Carbon dioxide adsorption experiment | Pore volume/cm$^3$·g$^{-1}$ | 0.0206 | 0.0271 | 0.0281 | 0.0244 | 0.0133 | 0.0128 | 0.0213 | 0.0133 |
| | Pore specific surface area/m$^2$·g$^{-1}$ | 132.856 | 168.463 | 153.955 | 141.948 | 94.137 | 85.012 | 115.587 | 102.615 |

The mercury injection experiments test data showed that the porosity of the eight low-rank coal samples ranged from 6.62% to 16.31%. The DLT coal sample had the largest porosity, and the RN coal sample was the smallest, with a porosity difference of 9.69%. The selected coal samples have a wide range of pore coverage. The average pore size ranged from 12.8 nm to 19.9 nm. The difference between BL coal sample with the largest average pore size, and HL coal sample with the smallest was 1.55 times. The difference between DLT coal sample with the maximum pore volume, and RN coal sample with the minimum value is 3.66 times. The difference between LSJ coal sample with the maximum specific surface area and RN coal sample with the minimum is 4.14 times.

The low-temperature $N_2$ adsorption experiments test data showed that the average pore size of the eight low-rank coal samples ranged from 4.56 nm to 11.08 nm. The average is small overall compared to the mercury injection experiment. The BL sample had the largest pore size, consistent with the results of the mercury injection test, and differed by a factor of 2.43 from DLT sample, which had the smallest average pore size. The specific surface area measured by BET method is smaller than that obtained by the mercury injection test, and the overall variation is greater. The difference between DLT sample with the largest specific surface area and the smallest BL coal sample was 12.39 times. The pore volumes measured by BJH method are also smaller than those obtained by the mercury

injection test. The difference between DLT sample with the largest pore volume and the smallest BL sample was 4.05 times.

The low-pressure $CO_2$ adsorption experiments test data showed that pore volume and specific surface optimum of low-rank coal microporous were calculated based on DFT model analysis. The distribution is consistent with the results of the mercury injection experimental tests. The DLT sample had the largest pore volume and RN sample had the smallest pore volume, with a difference of 2.20 times. The LSJ coal sample has the maximum specific surface area, and RN coal sample has the least specific surface area, with a difference of 1.98 times. The variation in pore volume and specific surface area of the micropore section varies less.

### 3.2. Joint Characterization of the Pore Structure of Full Aperture of Low-Rank Coal

Due to the different test principles and ranges of the three pore structure tests, it may have some errors in the test results for the same pore size. Using the experimental data above, the least difference or overlap in test results for the same pore size range was selected to connect and characterize the pore structure of the full aperture of the low-rank coal. The position of the pore joint should be within the test range of the corresponding test method. The three experimental data were bridged at 1.46~1.66 nm and 45.47~46.06 nm to plot the distribution of pore volume and pore specific surface area (Figures 7 and 8).

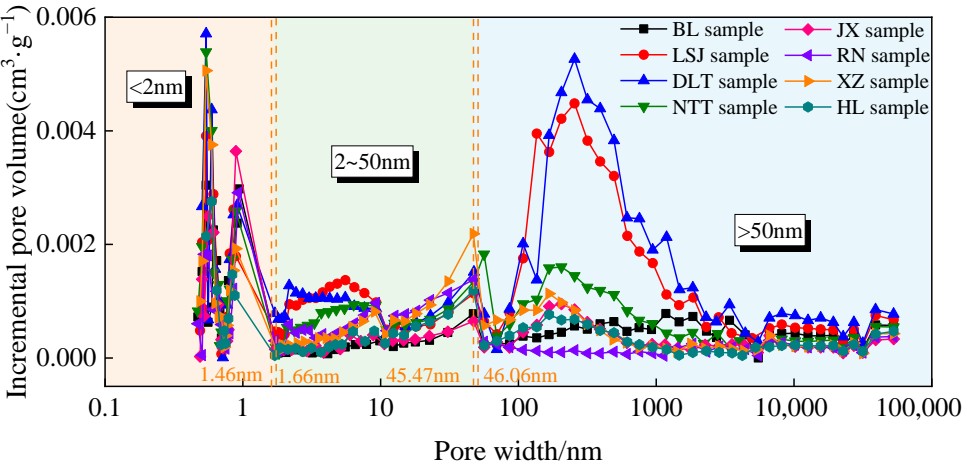

**Figure 7.** Pore volume distribution feature of full aperture.

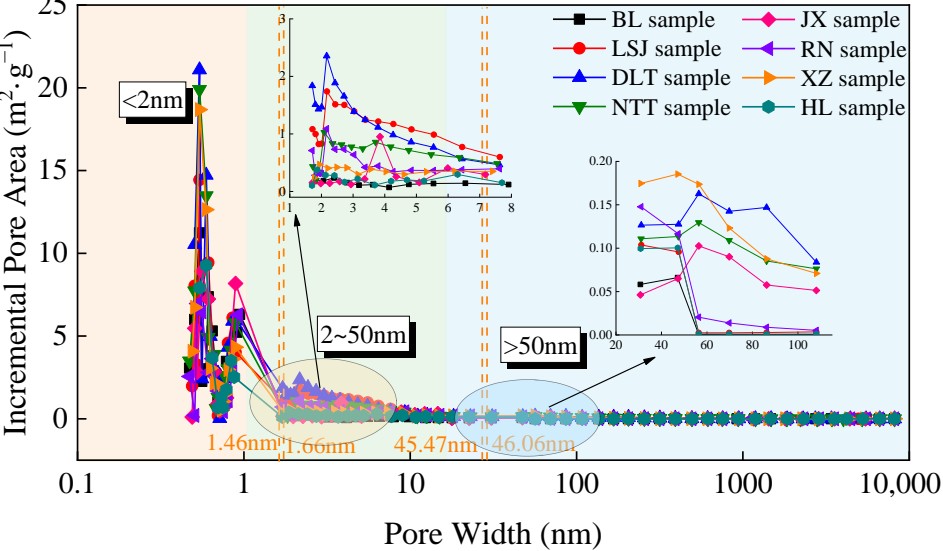

**Figure 8.** Specific surface distribution feature of full aperture.

From Figure 7, the pore space developed in the low-rank coal is mainly concentrated in the microporous stage. In the microporous section, the increase in pore volume was greater between 0.5 and 0.7 nm, and it reached a maximum around 0.55 nm. After 0.55 nm, the pore volume gradually decreases and shows a multi-peaked distribution. Small fluctuations in pore volume variation in the mesoporous section, with a tendency for the pore volume to increase as the pore size increases. The low-rank coal in the large pore section shows a trend of increasing and then decreasing between pore sizes of 100 and 1000 nm, with pore volumes decreasing gradually and fluctuating less after greater than 1000 nm.

From Figure 8, The specific surface area development of low-rank coal is mainly concentrated in the microporous stage. It had a large difference compared with the mesoporous and macropore stages. The peak of the microporous phase occurs around 0.55 nm, after which the specific surface area gradually decreases and shows a multi-peak distribution. The specific surface area of the mesoporous and macropore stages tends to decrease more steadily with the increase in pore size.

The pore structure characteristics are shown in Table 4. The specific surface area of the eight coal samples is mainly controlled by micropores. This can also be seen from Figure 8. The proportion of micropores reached 79.73–96.56%. The change rule of pore volume ratio is more complicated. The largest proportion of micropores is JX coal sample, accounting for 53.10%. The largest proportion of mesopores is RN coal sample, accounting for 46.81%. The largest proportion of macropores is DLT coal sample, accounting for 51.81%. As the degree of metamorphism increases, the coal is subjected to different stages of coalification. The evolution of pore space in different pore sizes appears to be significantly different due to the influence. Pore volume is more significantly influenced by coalification. The pore-specific surface area also has the same evolutionary characteristics as the pore volume. As the trends in specific surface area in the full aperture are mainly controlled by the microporous, the magnitude of change in the mesopore and macropore sections of the data is its slightest. The change rule is obvious. The change in pore volume with the increase in pore size is more complicated. Thus, this discussion focuses on the trends in the pore volume of low-rank coal.

**Table 4.** Pore structure characteristics of full aperture section.

| Sample ID | Pore Volume/cm$^3$·g$^{-1}$ | Proportion of Pore Volume/% | | | Specific Surface Area/m$^2$·g$^{-1}$ | Proportion of Specific Surface Area/% | | |
|---|---|---|---|---|---|---|---|---|
| | | $V_{mic}$ | $V_{mes}$ | $V_{mac}$ | | $S_{mic}$ | $S_{mes}$ | $S_{mac}$ |
| BL | 0.0391 | 52.36 | 10.26 | 37.39 | 63.6561 | 96.56 | 2.99 | 0.46 |
| LSJ | 0.0839 | 32.00 | 22.08 | 46.80 | 100.6521 | 81.02 | 15.29 | 3.69 |
| DLT | 0.0963 | 28.70 | 19.48 | 51.81 | 106.0858 | 79.73 | 15.22 | 5.04 |
| NTT | 0.0522 | 44.93 | 26.96 | 28.11 | 85.8990 | 86.26 | 11.26 | 2.49 |
| JX | 0.0257 | 53.10 | 19.31 | 27.59 | 42.9067 | 94.68 | 0.01 | 5.31 |
| RN | 0.0274 | 47.56 | 46.81 | 5.64 | 45.8984 | 82.71 | 17.13 | 0.17 |
| XZ | 0.0478 | 41.94 | 23.99 | 34.07 | 70.9783 | 90.25 | 7.33 | 2.42 |
| HL | 0.0252 | 41.85 | 27.13 | 31.02 | 35.7180 | 89.05 | 8.56 | 2.39 |

$V_{mic}$ is volume of micropore, $V_{mes}$ is volume of mesopore, $V_{mac}$ is volume of macropore. $S_{mic}$ is the specific surface area of micropores, $S_{mes}$ is the specific surface area of mesopores, and $S_{mac}$ is the specific surface area of macropore.

The microporous in low-rank coal were smaller pore formed by the stacking of macromolecular structures in the coal. In addition, there would exist some interlayer pore. The process of microporous evolution has similar characteristics to mesopore. The fracture of functional groups, branched chains, etc., in the macromolecular structure of the coal produces gas and forms some of the microporous. During this process, the increase in aromatic ring sheet layers of the microporous pore caused the aromatic sheet layers to stack up against each other as interlayer pore increases and thus the volume of the microporous pore increases.

The evolution of the mesopore is mainly controlled by the stacking of macromolecules such as coal molecular chains and aromatic ring lamellae. As the coalification process proceeds, the aromatization of the coal gradually increases and the condensation of the aromatic thick ring system further increases. At higher levels of metamorphism, some branched chains in coal are synthesized into aromatic rings and reduced. The macromolecular structure is more compact, and the molecular spacing is reduced, resulting in a reduction in mesopore pore volume. The pore volume of low-rank coal mesopore evolves in a similar trend to that of large pore with coalification but changes more smoothly during the coalification jump stage.

The change in macropore has the most significant effect on the change with pore volume in the full aperture. Excluding the effect of coalification jump stage on pore volume, the pore volume tends to increase and then decrease as the degree of metamorphism increases. The turning point of the change occurs at $R_{max}$ = 0.50%, when the first coalification jump begins. With the end of the coalification jump and the onset of the cracking reaction in the coal body, the volume of macropore pore space begins to decline.

### 3.3. Fractal Dimension Calculation of Coal Pore

Low-rank coal has strong heterogeneity and complex pore structure. Fractal geometric characteristics could be used to study the irregularity of pore structure and surface and to characterize the adsorption capacity of low-rank coal. The pore fractal dimension obtained by mercury intrusion experimental data was calculated by thermodynamic model [56]. The Frenkel–Halsey–Hill (FHH) model was used to calculate the pore fractal dimension obtained from the experimental data of low-temperature $N_2$ adsorption [57]. Combined with the data obtained by fitting the two models, the comprehensive fractal dimension is calculated to characterize the complexity of the pore structure of low-rank coal.

#### 3.3.1. Fractal Dimension Based on a Thermodynamic Model

During the mercury injection process, the amount of incoming mercury gradually increases as the pressure increases, resulting in constant increase in the surface energy of the pore [58]. The increase in the surface energy of mercury entering the pore is equal to the work exerted on the mercury by the external environment. The incremental pore surface energy in the whole system is consistent with the work done by the surroundings, thus giving the following Equation (2)

$$dW_n = \sigma \cos \alpha \, dS \tag{2}$$

where $W_n$ is the surface energy (J), $\sigma$ is the mercury surface tension (0.48 N/m), $\alpha$ is the contact angle between the mercury and the solid surface (130°), and $S$ is the specific surface area of the pore ($m^2$).

After the correction of Equation (2), the following equation was obtained:

$$\ln \frac{W_n}{r_n^2} = D_f \ln \frac{V_n^{1/3}}{r_n} + \ln C \tag{3}$$

where $r_n$ is the pore diameter (nm), $V_n$ is the pore volume ($m^3$), and $D_f$ is the fractal dimension of the pore surface area.

According to Equation (3), the fractal dimension was calculated for low-rank coal samples with pore size greater than 50 nm, and a fitting model of thermodynamic fractal was obtained (Figure 9).

The fitting results are shown in Table 5.

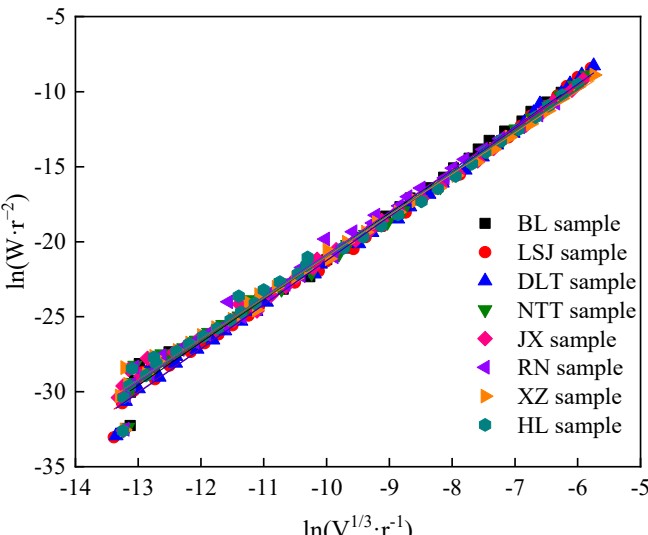

**Figure 9.** Thermodynamic model fitting curve.

**Table 5.** Fractal dimension that is based on the thermodynamic model.

| Sample | Fitting Equations | $R_1{}^2$ | $D_1$ |
|--------|-------------------|-----------|-------|
| BL | $y = 2.8737x + 7.7737$ | 0.9925 | 2.8737 |
| LSJ | $y = 2.9219x + 7.9515$ | 0.9953 | 2.9219 |
| DLT | $y = 2.9397x + 8.1395$ | 0.9953 | 2.9397 |
| NTT | $y = 2.7982x + 6.9738$ | 0.9926 | 2.7982 |
| JX | $y = 2.7646x + 6.6578$ | 0.9919 | 2.7646 |
| RN | $y = 2.8261x + 7.4171$ | 0.9894 | 2.8261 |
| XZ | $y = 2.7474x + 6.4992$ | 0.9918 | 2.7474 |
| HL | $y = 2.7631x + 6.6941$ | 0.9907 | 2.7631 |

### 3.3.2. Fractal Dimension Based on FHH Model

The FHH model, first proposed by Frenkel, Halsey and Hill, describes the theory of multilayer adsorption of gas molecules in porous media and is relatively simple to calculate [59]. On a fractal surface in a capillary condensation region and non-homogeneous porous solids, Avnir established the FHH equation of gas adsorption theory by studying the adsorption of gas molecules [60].

$$\ln V = C + (D_h - 3) \ln[\ln(\frac{P_0}{P})] \tag{4}$$

where $V$ is the amount of gas adsorbed at relative air pressure ($cm^3/g$). $P_0/P$ is the relative pressure, $C$ is a constant, and $D_h$ is the value of the fractal dimension of the porous material.

According to Equation (4), the fractal dimension of low-rank coal samples with meso-porous was fitted by the FHH model, shown in Figure 10. Fitting equation and fractal dimension as shown in Table 6. At $\ln[\ln(P_0/P)]$, the value of $\ln V$ changed significantly, so piecewise fitting fractal dimension values are presented here. The fitting degrees were high, which above 0.93. The fractal dimension ranged from 2.4794 to 2.8123, with an average of 2.6656. The fractal characteristics were obvious, and the pore structure of this section was highly complex.

The fractal dimension of low-rank coal samples with microporous was also fitted by the FHH model, as shown in Figure 11.

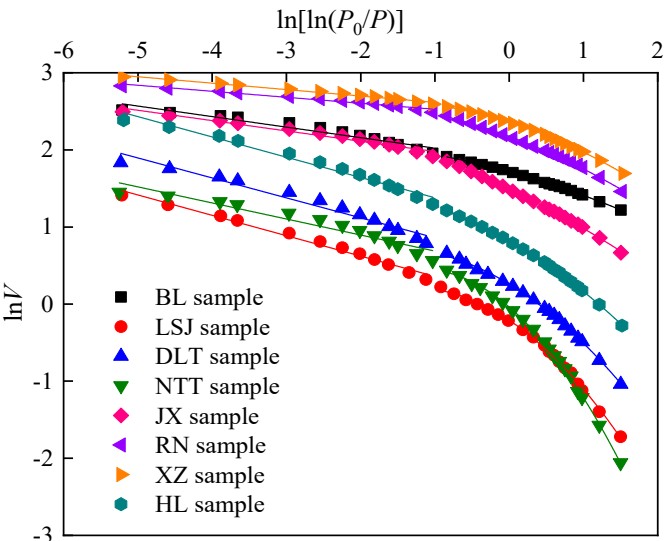

**Figure 10.** FHH model fitting curve of the mesoporous.

**Table 6.** Fractal dimension of the mesoporous based on FHH model.

| Sample | Relative Pressure < 0.7 | | | Relative Pressure > 0.7 | | | $D_2$ |
|---|---|---|---|---|---|---|---|
| | Fitting Equations | $R_{21}^2$ | $D_{21}$ | Fitting Equations | $R_{22}^2$ | $D_{22}$ | |
| BL | $y = -0.1367x + 1.8853$ | 0.9367 | 2.8633 | $y = -0.2832x + 1.6999$ | 0.9889 | 2.7168 | 2.8066 |
| LSJ | $y = -0.2642x + 0.0942$ | 0.9862 | 2.7358 | $y = -0.7645x - 0.3073$ | 0.9526 | 2.2355 | 2.5326 |
| DLT | $y = -0.2589x + 0.6009$ | 0.9615 | 2.7411 | $y = -0.6817x + 0.1944$ | 0.9688 | 2.3183 | 2.5694 |
| NTT | $y = -0.2082x + 0.4790$ | 0.9326 | 2.7918 | $y = -0.9773x - 0.1844$ | 0.9474 | 2.0227 | 2.4794 |
| JX | $y = -0.1339x + 1.8445$ | 0.9673 | 2.8661 | $y = -0.4811x + 1.4708$ | 0.9941 | 2.5189 | 2.7251 |
| RN | $y = -0.0779x + 2.4487$ | 0.9686 | 2.9221 | $y = -0.3852x + 2.1473$ | 0.9854 | 2.6148 | 2.7973 |
| XZ | $y = -0.0833x + 2.5326$ | 0.9872 | 2.9167 | $y = -0.3402x + 2.3229$ | 0.9713 | 2.6598 | 2.8123 |
| HL | $y = -0.2632x + 1.1156$ | 0.9772 | 2.7368 | $y = -0.5951x + 0.7743$ | 0.9836 | 2.4049 | 2.6020 |

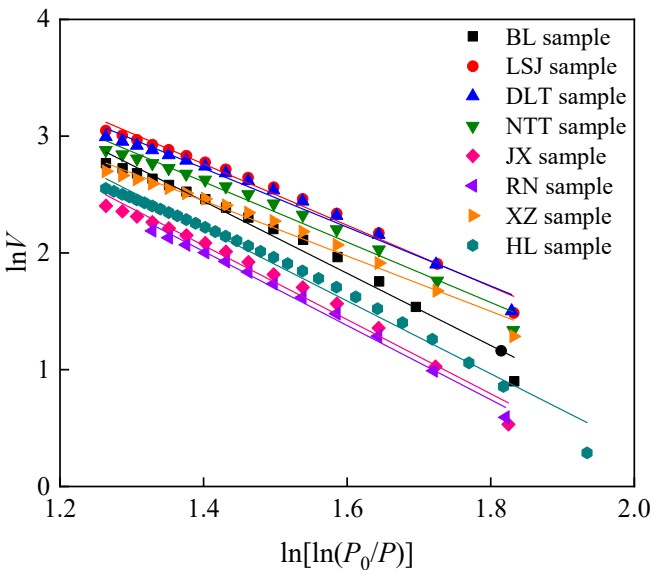

**Figure 11.** FHH model fitting curve of the microporous.

Fitting equation and fractal dimension are shown in Table 7. The fitting degrees of micropore section were high, which was above 0.98. The fractal dimension is 1.0084 to 1.8414, and the average value is 1.4107.

**Table 7.** Fractal dimension of the microporous based on FHH model.

| Sample | Fitting Equations | $R_3{}^2$ | $D_3$ |
|--------|-------------------|-----------|-------|
| BL | $y = -2.7544x + 6.3004$ | 0.9894 | 1.2456 |
| LSJ | $y = -2.4086x + 6.1353$ | 0.9911 | 1.5914 |
| DLT | $y = -2.3040x + 5.9513$ | 0.9889 | 1.6960 |
| NTT | $y = -2.3447x + 5.8928$ | 0.9872 | 1.6554 |
| JX | $y = -2.8999x + 6.1255$ | 0.9893 | 1.1001 |
| RN | $y = -2.9962x + 6.2010$ | 0.9951 | 1.0084 |
| XZ | $y = -2.1586x + 5.4703$ | 0.9889 | 1.8414 |
| HL | $y = -2.8524x + 6.2028$ | 0.9942 | 1.1476 |

*3.4. Relationship between the Fractal Dimension of Low-Rank Coal and the Degree of Coal Metamorphism*

The calculated fractal dimensions were concentrated between 2.4 and 3.0, and the correlation degree is 86.09–98.30%, with high correlation and good fractal characteristics. The fractal dimension of RN sample joint pore greater than 50 nm pore size segment was greater than 3. It had been suggested that there may be compressive damage to the fractures and pore in the coal as a result of high-pressure mercury injection. It is also possible that the coal seam is highly metamorphosed or highly fractured and deformed. The high-pressure stage (>10 MPa) and pore sizes greater than 50 nm have been corrected for compression. It was unlikely that the first cause of the fractal dimension was greater than 3. In conjunction with the above experimental analysis, possible due to a high degree of fracture deformation in RN coal sample.

As a porous, non-homogeneous solid, coal has different fractal characteristics at different pore-size sections. In order to better characterize quantitatively the complexity of the experimental coal samples and their rough surface, the comprehensive fractal dimension was calculated. The comprehensive fractal dimension was obtained by weighing the pore volume ratios of different pore size sections as weights and summing the fractal dimensions of different pore size sections.

$$D_{\mathrm{t}} = \sum D_i \, T_i \tag{5}$$

where $D_{\mathrm{t}}$ is the integrated fractal dimension of the coal, $D_i$ is the fractal dimension corresponding to the $i$ pore size section, $T_i$ s the pore volume ratio corresponding to the $i$ pore size section (%), and $i$ is the $i$ Pore Size Section and is a positive integer.

The pore volume share of microporous, mesopore and macropore was calculated. Based on the range of applications of the fractal model discussed above, the integrated fractal dimension is calculated for the full aperture section. The integrated fractal dimension of the full aperture section of the coal sample is calculated from Equation (5). The results are shown in Table 8.

**Table 8.** Comprehensive fractal dimensions of full aperture.

| Sample | Microporous Stage | | | Mesopore Stage | | | Macropore Stage | | | Synthesis of Fractal Dimensions |
|--------|-------------------|--------|--------|----------------|--------|--------|-----------------|--------|--------|----------------------------------|
| | $R_1{}^2$ | $D_1$ | $V_1$/% | $R_2{}^2$ | $D_2$ | $V_2$/% | $R_3{}^2$ | $D_3$ | $V_3$/% | |
| BL | 0.9894 | 1.2456 | 52 | 0.9569 | 2.8066 | 10 | 0.9925 | 2.8737 | 38 | 1.9916 |
| LSJ | 0.9911 | 1.5914 | 32 | 0.9726 | 2.5326 | 21 | 0.9953 | 2.9219 | 47 | 2.4397 |
| DLT | 0.9889 | 1.6960 | 29 | 0.9645 | 2.5694 | 19 | 0.9953 | 2.9397 | 52 | 2.5087 |
| NTT | 0.9872 | 1.6554 | 45 | 0.9386 | 2.4794 | 27 | 0.9926 | 2.7982 | 28 | 2.1979 |
| JX | 0.9893 | 1.1001 | 53 | 0.9782 | 2.7251 | 19 | 0.9919 | 2.7646 | 28 | 1.8749 |
| RN | 0.9951 | 1.0084 | 48 | 0.9754 | 2.7973 | 46 | 0.9894 | 2.8261 | 6 | 1.9404 |
| XZ | 0.9889 | 1.8414 | 42 | 0.9807 | 2.8123 | 24 | 0.9918 | 2.7474 | 34 | 2.3825 |
| HL | 0.9942 | 1.1476 | 42 | 0.9798 | 2.6020 | 27 | 0.9907 | 2.7631 | 31 | 2.0411 |

During coal formation, coalification does not evolve linearly but undergoes several jumps. Low-rank coal pore is subject to more complex changes by coalification [61,62]. The aliphatic, alicyclic functional groups, and side chains are shed from the aromatic layer to

form methane-based volatiles when the coalification was at a $R_{max}$ of 0.50% to 0.60% [63]. The asphaltene caused by asphalting shows that the first jump begins.

During the first coalification jump, the pore structure is controlled by the dissociation and polymerisation of functional groups and aromatic structure, and the trend of change fluctuates considerably. The phenomenon is most evident in the microporous stage. Here, it is shown that, from $R_{max}$ = 0.50%, a substantial reduction in pore volume and specific surface area occurs, and the comprehensive fractal characteristics of the full aperture diminish sharply (Figure 12).

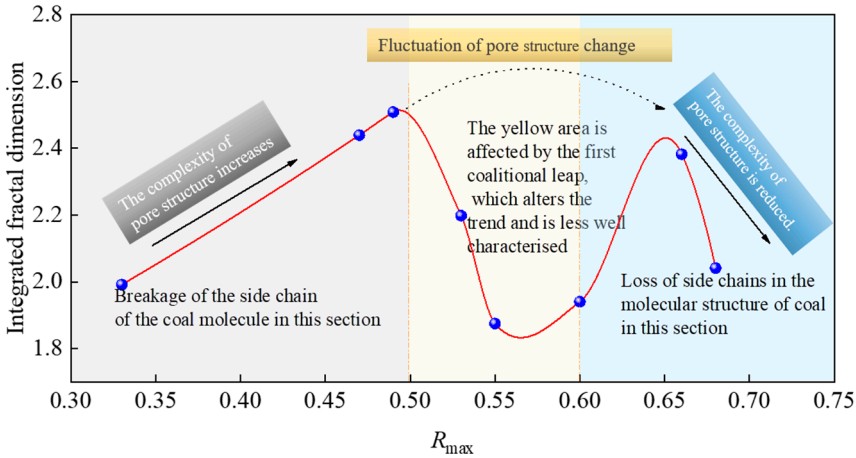

**Figure 12.** Trends in integrated fractal dimensionality with coal evolution.

As the coalification of low-rank coal increases, the evolution of the pore structure of low-rank coal can be broadly divided into two stages, excluding the coalification jump stage. The first stage is before the start of the first coalification jump ($R_{max}$ < 0.30%), The chemical reactions in coal are dominated by the formation of hydrocarbons, the destruction of coal molecular chains and aromatic ring lamellae, the gradual increase in pore volume, and the increase in specific surface area of coal. After the end of the first coalification jump, the coalification reaction gradually changed to a predominantly cracking reaction. At this stage, moisture, volatile matter, hydrogen, and oxygen content gradually decrease. The organic molecules are gradually arranged in a regular manner. Furthermore, as the degree of polymerisation increases, the carbon content gradually increases. The side chains of the coal molecules decrease, and the layer space decreases, resulting in a gradual decrease in the pore volume and specific surface area of the coal.

In order to study the development and complexity of the pore volume and specific surface area of low-rank coal, the pore volume and specific surface area variation graphs were obtained, as shown in Figures 13 and 14.

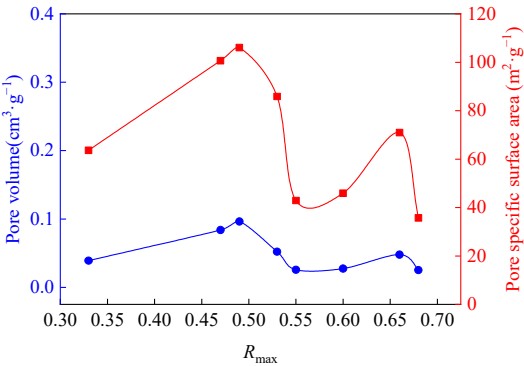

**Figure 13.** Trend in pore volume and specific surface area with coal evolution.

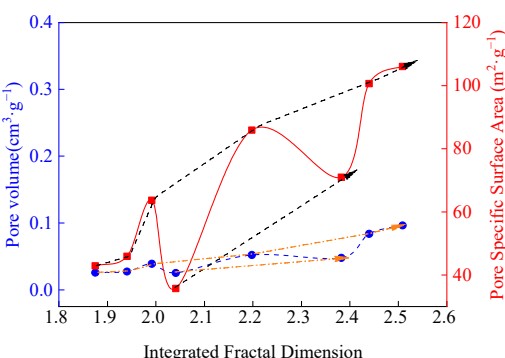

**Figure 14.** Trend in pore volume and specific surface area with integrated fractal dimensionality.

From Figure 13, the trends in pore volume and specific surface area with coal evolution are similar to those in integrated fractal dimensionality with coal evolution. The trend in specific surface area coincides well with the trend in integrated fractal dimension with an increasing degree of metamorphism. As the specific surface area is mainly controlled by the microporous, the complexity of the pore structure depends more on the microporous.

From Figure 14, there is an overall trend of increasing pore volume and specific surface area as the number of integrated fractal dimensions increases. The larger the pore volume or specific surface area, the more complex the pore structure is. The pore volume and specific surface area before the completion of the coalification jump are larger than those after the completion of the coalification jump. In these two stages, the pore volume and specific surface area increase with the increase of pore structure complexity.

The study shows that the change in specific surface area with $R_{max}$ is most consistent with the change in total fractal dimension with $R_{max}$. To verify the accuracy of the experimental law, some experimental data in the literature are cited, and the results are basically consistent with the law obtained in this paper (Figure 15). The law of pore structure variation with coalification mentioned in this paper can be confirmed by subsequent research.

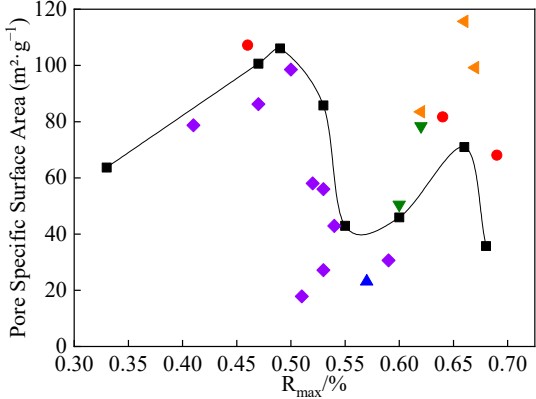

**Figure 15.** Trend in specific surface area with coal evolution [63–67].

## 4. Conclusions

In mercury injection experiments, the pore volume and specific surface area of low-rank coal depend mainly on the mesopore. Coal samples with $R_{max}$ between 0.40% and 0.55% also have a large number of developed pores in the large pore section. In low-temperature $N_2$ adsorption experiments, it shows that pore volume and specific surface area were more developed in 2 nm to 50 nm mesopore section. In $CO_2$ adsorption experiments, pore volume and specific surface area of low-rank coal depend on the microporous and show an "increasing–decreasing–increasing" trend at 0.4 nm–0.55 nm–0.7 nm–0.9 nm.

The pore structure characteristics of full aperture were characterized using low-pressure $CO_2$ adsorption to characterize the microporous, low-temperature $N_2$ adsorption,

and mercury compression. It is bridging at 1.46–1.66 nm and 45.47–46.06 nm, respectively. The pore-specific surface area of the full aperture was mainly controlled by micropores. The specific surface area of micropores accounts for 79.73–96.56% of the full aperture. The change in pore volume is more complicated due to the influence of coalification jump. The change rule needs specific analysis.

The pore volume of the full aperture of low-rank coal is mainly controlled by the macropore via the segmental union pore. The specific surface area is mainly controlled by micropores, which are more effectively controlled. The pore fractal characteristics of full aperture in low-rank coal vary between the different degrees of metamorphism, with the fractal dimension ranging from 1.8749 to 2.5087. The macropore fractal features are most pronounced, with fractal dimensions ranging from 2.7474 to 2.9397. Mesopores are next in line, with fractal dimensions ranging from 2.4794 to 2.8123. The micropores are most affected by coalification and have the weakest fractal characteristics, with fractal dimensions of 1.0084 to 1.8414.

The fractal characteristics of low-order coal fluctuate in $R_{max}$ = 0.50–0.60% stage as the degree of metamorphism increases by the coalification jump. In $R_{max}$ =0.30–0.50% stage, the pore structure complexity of low-rank coal increases with the vitrinite reflectivity. After the first coalification jump, the complexity of the pore structure in low-rank coal decreases with the vitrinite reflectivity.

**Author Contributions:** Investigation, Data curation, Writing-original draft preparation, Formal analysis, Y.Z.; Conceptualization, Methodology, Validation, funding acquisition, S.L.; Methodology, Resources, Y.B.; Methodology, Validation, Supervision, H.L.; Data curation, Formal analysis, Resources, Y.C.; supervision, Writing—review & editing, J.Z. All authors have read and agreed to the published version of the manuscript.

**Funding:** This research was funded by National Natural Science Foundation of China, (grant No. 52074217, funder: Li S). China Postdoctoral Science Foundation (grant No. 2022MD713795, funder: Bai Y). Natural Science Foundation of Shaanxi Province (grant No. 2022JQ-347, funder: Bai Y). Shaanxi Provincial Department of Education Youth Innovation team (grant No. 22JP048, funder: Bai Y). The authors would like to thank these founda-tions for the financial support.

**Institutional Review Board Statement:** Not applicable.

**Informed Consent Statement:** Not applicable.

**Data Availability Statement:** The experimental data can be provided upon request to the corresponding author.

**Conflicts of Interest:** The authors declare no conflict of interest.

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
