# Peer review of "Joint Characterization and Fractal Laws of Pore Structure in Low-Rank Coal"

_sustainability, doi:10.3390/su15129599_

Round 1

Reviewer 1 Report

This manucript focus on the pore structure in low-rank coal. It is well written and some useful data has been given. It can be accepted. Some comments are as followed.

1 Samples with different geological backgrounds should be given attention.

2 Trends in integrated fractal dimensionality with coal evolution. The Ro value is almostly same. You should find some data of high coal rank to compare them togather. 

2 Trends in integrated fractal dimensionality with coal evolution. The Ro value is almostly same. You should find some data of high coal rank to compare them togather. 

Reviewer 2 Report

The article is very interesting, coal seams are a hot topic, as is alternative energy. But there are a few notes on design and statistical analysis:

1. The index of literature used in the article is not designed in accordance with the standard: [1], [2-5];

2. Words in bold in line 38 are deleted;

3. The number 78 is written in line 36, what does this mean? Doesn't make sense. There are 52 links in total.

4. Must be displayed in numerical order. You have 1, 24, 5, 6, etc.

5. How many times were the results of Table 1,2,3,4,5,6,7 repeated? Is this the result of one try? If once, why? As you know, for the reliability of the results, experiments must be carried out at least 3 times. If several times, where are the statistics?

6. Fig. 2 (a, b) should be enlarged, the experimental system is not clear. Especially (a).

7. Do you have data on scanning electron microscopes? You must add.

8. In general, the article lacks statistical analysis. You need to work on this. It is important.

Reviewer 3 Report

Pore structure of coal is the basis for further study of adsorption, diffusion and percolation of coalbed methane. This study has carried out a more complete and targeted study on the pore structure characteristics of low rank coal. In contrast to conventional pore structure studied, the author considered the effect of coal formation on pore variation. The thesis had explicit objectives, clear ideas, and solid conclusions. The following issues need to be refined before publication:
1.
The final two sentences of the abstract lack clarity in their logic and fail to convey a clear regulation. They need to be rephrased for a more concise summary.

2. Standardizing the description of the "low-temperature nitrogen adsorption experiment" and the "low-pressure CO2 experiment" throughout the article.

3. The titles in the text should follow a consistent capitalization style, either with only the first letter of the sentence capitalized or with the first letter of each word capitalized.

4. Why were the experimental data of the eight coal samples depicted separately on two different graphs?

5. Please simplify the introduction of the fractal model as it is not the main focus of the discussion.

6. Please ensure that percentages in both the table and the article are consistently rounded to two decimal places.

7. The author needs to explain in detail in the "Introduction" what is the effects of Pore structure of coal. Moreover, the significance and progress of your research needs to be explained in "introduction" section. Some references should be cited as follows: (a) Optimization of gob ventilation boreholes design in longwall mining. International Journal of Mining Science and Technology 2020;30(6):811-817. (b) Investigation of collector mixtures on the flotation dynamics of low-rank coal[J]. Fuel, 2022,327: 125171.

8. The “Conclusions” and “Introduction” sections should be refined and shorten.

9. The quality of the figures should be improved.

10. Authors should carefully check the format of references and citations.

no.

Reviewer 4 Report

The article I received for review entitled: "Joint Characterization and Fractal Laws of Pore Structure in Low-rank Coal" I find very interesting, well-written, and legible. The subject of coal, despite the fact that it has been studied for years, still carries some unknowns, and the complicated structure of this mineral causes ambiguity in its opinion and assessment. In the article, the authors focused on low-rank coal. It is good that they tested 8 coal samples, although due to the variable characteristics of this mineral, it would be good to include more samples in subsequent articles and to enrich the research with the percentage analysis of maceral groups. The maceral composition (mainly the ratio of vitrinite to inertinite) has a large impact on the character of the pores. Vitrinite is characterized by the presence of micropores, while macerals of the inertinite group have a lot of macropores. Therefore, in order to compare coals with a similar degree of coalification, it would also be good to compare the percentage of content of individual groups of macerals.
Note on Table 1. Sample BL has a Ro of 0.33%. Such low coalification indicates that it is lignite, the others belong to low-rank hard coal. However, one result seems unlikely to me - Vdaf is surprisingly low. Such a small presence of volatiles is assigned to coals with a much higher degree of coalification. Please check if there is any measurement error.

I did not find a citation of Table 6 in the text.

Literature in the text is quoted in an illegible way, please format it in accordance with the requirements of the publishing house (e.g. [23], and in the text, it is without parentheses, which is very illegible)

The article is well written and valuable and is suitable for publication in the journal Sustainability, and my comments (except for the last two) are rather suggestions that the authors can use in this or subsequent articles

Reviewer 5 Report

The authors' article is devoted to an important and topical issue related to the study of the porous structure of low-grade coals.

The gas factor has a decisive influence on the safety of mining operations in coal mines. Currently, in the main existing coal basins in the world, methane-hazardous mines of category III, super-category and outburst mines account for about 60%.

Abundant methane release from mined highly gas-bearing coal seams hinders the ability of coal mining equipment in terms of gas factor, which significantly affects the profitability of coal mines. In the conditions of mining methane-bearing coal seams, the full realization of the capabilities of modern coal mining equipment is possible only if the coal seams are degassed in advance by industrial methane production in combination with effective methods of controlling gas release into mine workings by means of degassing and ventilation.

Thus, in modern conditions, the extraction of coal-bed methane is becoming the main factor in ensuring safe working conditions, increasing its productivity, and one of the accompanying factors - ensuring the socio-economic development of various regions in the world through the use of coal-bed methane.

The studies presented in the paper are undoubtedly of interest to readers in the field under consideration.

 However, it would be necessary to clarify a number of comments that are available to the article:

 1. In the introduction, the literature review should be expanded by mentioning the dangers that are possible in coal mines, in particular with methane explosions and various aerological risks:

- Bosikov I.I., Martyushev N.V., Klyuev R.V., Savchenko I.A., Kukartsev V.V., Kukartsev V.A., Tynchenko Y.A. Modeling and complex analysis of the topology parameters of ventilation networks when ensuring fire safety while developing coal and gas deposits. Fire 2023, 6, 95. https://doi.org/10.3390/fire6030095.

- Balovtsev S.V., Skopintseva O.V., Kolikov K.S. Aerological risk management in preparation for mining of coal mines. Sustainable Development of Mountain Territories. 2022;14(1):107-116. https://doi.org/10.21177/1998-4502-2022-14-1-107-116.

2. Has the measurement results in Table 1 been compared with similar results in other regions of the world that have coal of similar composition with samples taken from mines.

3. The letter designations in Figure 2 should be made more legible and the scheme of the test system operation should be explained in more detail.

4. In the "Materials and Methods" section, one should dwell in more detail on the methods used in the work.

5. Based on the dependencies shown in Figures 3-6, a regression analysis should be carried out and the corresponding approximation equations with determination coefficients should be presented for further use in calculating and predicting output parameters.

6. It would be necessary to systematize the results presented in figures 7, 8, since, in my opinion, in the course of their analysis, quite important conclusions were made, which could be given in a generalized table.

7. The article should provide a more detailed analysis of Figure 15.

8. It is not clear from the article whether it is planned to obtain a patent for the applied research methodology and what are the prospects for its use in similar conditions in other regions of the world?

Reviewer 6 Report

Dear authors,

Your manuscript is very interesting and you really represent a useful research for a wide scientific and professional public.

I have certain objections, which should be taken into account:

1. Figure 3. All adsorption curves should be inserted into one diagram

2. State: : All eight low-rank coal samples shown significant mercury injection hysteresis loops, indicating that the open-pore spaces are more developed and  the inter-pore connectivity is better. And further, you conclude: Mercury output efficiency is calculated at 30.85%-48.34%. The mercury release efficiency is medium, which indicates that there are both open and semi-open pores in the coal samples, and the connectivity of the pores is good.

- How would you explain the similarity of the adsorption curves for example BL and NTT samples - NTT contains far more moisture and BL much more ash? Although both samples contain significant amounts of fixed carbon. The parameter Rmax /% differs.

3. Does an increase in pressure cause deformation of the pores of prepared samples without adsorbent (mercury, nitrogen, carbon dioxide)?

4. After modification and linear regression, if you look now at the diagram in Figure 4, you see a big difference in the characteristics on the ordinate for the BL and NTT samples compared to Figure 3. Explain that.

5. I would put pictures 9, 10 and 11 in one diagram, each separately. The fitting curve of the thermodynamic model was not presented in the manuscript.

I wish you luck in your scientific research work.

Best Regards

Round 2

Reviewer 2 Report

Thank you for your work, very important topic. All remarks are exhausted.

Reviewer 3 Report

The paper has been well modified and can be accepted.

no.

Reviewer 5 Report

The authors answered the questions posed. I recommend for publication in the presented form.